# GC-MS Analysis of the Essential Oil from *Seseli mairei* H. Wolff (Apiaceae) Roots and Their Nematicidal Activity

**DOI:** 10.3390/molecules28052205

**Published:** 2023-02-27

**Authors:** Shengli Shi, Xinsha Zhang, Xianbin Liu, Zhen Chen, Hewen Tang, Dongbao Hu, Hongmei Li

**Affiliations:** 1School of Chemistry, Biology and Environment, Yuxi Normal University, Yuxi 653100, China; 2Faculty Affairs Center, Yuxi Normal University, Yuxi 653100, China

**Keywords:** *Seseli mairei* H. Wolff, *Bursaphelenchus xylophilus*, falcarinol

## Abstract

The essential oil (EO) was extracted from aerial parts with insecticidal and fungicidal activity. Herein, the hydro-distilled essential oils of *Seseli mairei* H. Wolff roots were determined by GC-MS. A total of 37 components were identified, (*E*)-beta-caryophyllene (10.49%), *β*-geranylgeranyl (6.64%), (*E*)-2-decenal (6.17%) and germacrene-D (4.28%). The essential oil of *Seseli mairei* H. Wolff had nematicidal toxicity against *Bursaphelenchus xylophilus* with a LC_50_ value of 53.45 μg/mL. The subsequent bioassay-guided investigation led to the isolation of three active constituents: falcarinol, (*E*)-2-decenal, and octanoic acid. The falcarinol demonstrated the strongest toxicity against *B. Xylophilus* (LC_50_ = 8.52 μg/mL). The octanoic acid and (*E*)-2-decenal also exhibited moderate toxicity against *B. xylophilus* (LC_50_ = 65.56 and 176.34 μg/mL, respectively). The LC_50_ of falcarinol for the toxicity of *B. xylophilus* was 7.7 and 21 times than that of octanoic acid and (*E*)-2-decenal, respectively. Our findings demonstrate that the essential oil from *Seseli mairei* H. Wolff roots and their isolates may be developed as a promising natural nematicide.

## 1. Introduction

Plant-parasitic nematodes (e.g., *Meloidogyne* spp., *Bursaphelenchus* spp., *Heterodera* and *Globodera* spp.) may cause severe damage to plants and crops. It is estimated that the annual loss in global crop production due to plant-parasitic nematodes is around 8.8–14.6% of crop yields [1]. The pine wood nematode (PWN), *Bursaphelenchus xylophilus,* can lead to pine wilt disease, and has resulted in $ 1 billion economic losses annually [1,2]. Since its discovery in Jiangsu province, China in 1982, PWN has been found in 14 provinces of China [3], which causes extensive damage in pine forests. In the last decades, these nematodes were controlled mainly by the overuse of chemically synthesized nematicides and soil fumigants. Heavily repeated application of these chemicals has resulted in many problems such as pest resurgence, resistance to nematicides, pesticide residues in plants, groundwater contamination, and fatality in non-target organisms [4]. Therefore, it is necessary to develop the less toxic activity pesticides. Phytochemicals found in a variety of plants show a great potential in controlling nematodes [5]. Essential oil (EO) is a natural volatile substance, with complex mixture of mainly terpenoids, was found to stand out for nematicidal toxicity [4,5,6,7,8,9,10,11]. According to previous literature, some EO extracted from diverse plants such as plants in *Boswellia carterii*, *Cymbopogon citrates*, *Eugenia caryophyllata*, etc., exhibited nematicidal activities against PWN [4,12].

*Seseli mairei* H. Wolff, a perennial herbaceous plant with erect stems growing 15 to 80 cm tall, known as “Zhu Ye Fang Feng”, is mainly distributed in Yunnan, Sichuan, Guizhou, and Guangxi provinces of China, as well as northern Thailand [13]. Its roots are brown cylinder with sweet taste, as herbal remedies for the treatment of inflammation, swelling, rheumatism, pain, and common cold in traditional Chinese medicine [14]. Previously phytochemical investigation on *Seseli mairei* H. Wolff has identified a number of coumarins, phenylpropanoids, triterpenoids and polyacetylenes [13,14,15,16,17,18]. At present, there are no reports on the volatile compounds and nematicidal activities of *Seseli mairei* H. Wolff roots. Therefore, we aimed to evaluate the chemical components and nematicidal activity of EO from *Seseli mairei* H. Wolff roots against PWN.

## 2. Results and Discussion

The 6-h steam distillation of *Seseli mairei* H. Wolff roots yielded 25 mL of EO (yellow) and the density of EO was 0.92 g/mL. Based on the GC and GC-MS analyses of the EO of *Seseli mairei* H. Wolff, 37 components were identified and quantified, which accounted for 97.84% of the total components (Table 1).

The major components in the EO were falcarinol (25.25%), octanoic acid (10.49%), *β*-caryophyllene (6.64%), (*E*)-2-decenal (6.17%) and germacrene-D (4.28%) (Table 1). The EO of *Seseli mairei* H. Wolff roots contained 51.22% terpenoids (monoterpenoids and sesquiterpenoids). Monoterpenoids contained 19 of the 37 components, accounting for 27.50% of the total EO, while 10 of the 37 components were found in sesquiterpenoids (23.72% of the crude EO). This study is the first to demonstrate the chemical compositions of EO in *Seseli mairei* H. Wolff roots. However, these findings are not similar with several EO isolated from *Seseli* species. For example, the EO derived from the *S. seseloides* aerial parts collected from Tianshui, Gansu province, China mainly contained myristicin (64.05%) and apiol (8.98%) [19]. Besides, polyalcyne falcarinol (38.8%), *n*-octanal (10.1%), (*E,E*)-2,4-decadienal (8.8%) and ar-curcumene (6.2%) were the components in the EO of *S. gracile* roots collected from Derdap gorge, Serbia in September 2013 [20], while Kurkcuoglu et al. [21] found that the main constituents in the EO extracted from the *S. gummiferum subsp. ilgazense* aerial parts collected from Kastamonu, Turkey were α-pinene (7.2%), germacrene D (9.5%) and sabinene (28.8%). However, EO of *S. rigidum* roots at flowering stage collected from Smolyan, Bulgaria contained elemol (8.7%), sabinene (12.4%), falcarinol (48.1%) and germacrene D [22], while Jovanovic et al., [23] reported that the main constituents of *S. pallasii* root (collected in August 2013 from Kravlje area, Eastern Serbia) EO were *n*-undecane (13.3%), (*Z*)-β-ocimene (34.5%) and *n*-nonane (45.2%) [23]. These examples show that due to various reasons in nature, the inherent genetic variation of plants or their external environmental changes, the appearance of biological species has changed, and the chemical components contained in them will change accordingly in composition and content.

The EO of *Seseli mairei* H. Wolff roots possessed nematicidal toxicity against the PWN (*B. xylophilus*) with a LC50 value of 53.45 μg/mL. The EO exhibited 23 times less toxicity than a positive control, rotenone (LC50 = 2.32 μg/mL) against *B. xylophilus*. However, when compared with the other EO in previous studies using the same bioassay, the EO of *Seseli mairei* H. Wolff roots exhibited stronger toxicity against the PWN, e.g., EO of *Ajowan* (*Trachyspermum ammi*, LC_50_ = 431 μg/mL), *Allspice* (*Pimenta dioica*, LC_50_ = 609 μg/mL) and *Litsea* (*Litsea cubeba*, LC_50_ = 504 μg/mL) [6]; EO of *Cymbopogon citratus* (LC_50_ = 0.456 μL/mL), *Origanum vulgare* (LC_50_ = 0.754 μL/mL), *Ruta graveolens* (LC_50_ = 0.232 μL/mL), *Satureja montana* (LC_50_ = 0.261 μL/mL), *Thymbra capitata* (LC_50_ = 0.265 μL/mL) and *Thymus caespititius* (LC_50_ = 0.972 μL/mL) [24]. The isolated compound, falcarinol (LC_50_ = 8.53 μg/mL) exhibited stronger nematicidal toxicity against *B. xylophilus* than the EO (Table 2).

However, two other isolates, (*E*)-2-decenal (LC_50_ = 176.34 μg/mL) and octanoic acid (LC_50_ = 65.56 μg/mL) showed weaker toxicity against the PWN (*B. xylophilus*) (Table 2). Thus, the nematicidal toxicity of the EO may be attributed to the active compound falcarinol. Compared with rotenone, falcarinol displayed only 4 times lesser toxic to the PWN. In the previous studies, falcarinol, (*E*)-2-decenal and octanoic acid were reported to exhibit nematicidal toxicity against plant-parasitic nematodes such as *B. xylophilus*, root-knot nematodes (*Meloidogyne javanica*, *M. incognita*) and cereal cyst nematode (*Heterodera avenae*) [25,26,27,28,29,30].

In summary, the nematicidal toxicity of the EO of *Seseli mairei* H. Wolff and the isolated compounds, especially falcarinol, is lesser compared with the chemical nematicides, which may be developed as potential natural nematicides for the control of plant-parasitic nematodes. Therefore, plant EO can play a vital role in the management of plant-parasitic nematodes and reduce the risks possessed by synthetic chemicals. Nevertheless, further studies on the practical application of EO as novel nematicides are needed to enhance the stability and efficacy as well as to improve cost effectiveness.

## 3. Materials and Methods

### 3.1. Plant Material and EO Extraction

The air-dried roots (30 kg) of *Seseli mairei* H. Wolff (collected in Dongchuan District, Kunming City, Yunnan Province, November 2021) were purchased from Yunnan Medicinal Material Co., Ltd. (No. 378, Wujin Road, Kunming 650041, Yunnan Province, China). Voucher specimen (no. 001-tssf-01657) was deposited at the museum of School of Chemistry, Biology and Environment, Yuxi Normal University, China. The plant specimens were chopped into small pieces. Each small piece (500 g) was added into 2000 mL of tap water and boiled in a Clevenger apparatus, followed by steam distillation for 6 h. Steam condensate (including EO) from distillation was harvested and placed in a flask. The EO was extracted with the same volume of n-hexane (a non-polar solvent with moderate boiling point) using a separation funnel. Evaporation of the solvent was conducted using a vacuum rotary evaporator. The samples were dried over anhydrous Na_2_SO_4_, and n-hexane was evaporated to obtain EO [10,27]. The EO was kept at 4 °C until further analysis.

### 3.2. Nematodes

The PWN (*B. xylophilus*) was extracted from chips of the infected pine wood harvested in Shuifu city, Yunnan province, China (28.63 °N latitude and 104.40 °E longitude, altitude 500 m) in September 2020, and extracted with the modified Baermann funnel method [17]. After rinsing 3 times with sterile distilled water (H_2_O), the PWN isolate was reared on *Botrytis cinerea* cultures. The gray mold fungus (*B. cinerea*) was cultured on potato dextrose agar (PDA) in a growth chamber at 27–29 °C in darkness. Subsequently, the plate was inoculated with *B. xylophilus* and maintained in the growth chamber at 27–29 °C in darkness until the fungal mycelium was fully digested by the PWN. After that, the PWN were harvested using the modified Baermann funnel method [31], rinsed 3 times with a mixture of 0.002% actinone and 0.1% streptomycin sulfate to eliminate any surface fungal or bacterial contaminants, and then employed for bioassays immediately.

### 3.3. Nematicidal Activity

Range-finding tests were performed to select the appropriate testing concentrations of EO and its isolates. Standard nematode suspension was prepared by dilution with sterilized H_2_O to obtain 100 juveniles/mL. Subsequently, 500 μL standard juvenile suspensions was introduced into a 24-well tissue culture plate. The number of active juveniles in each well was counted using a stereoscope at 5× and 10× before the addition of 500 μL stock solution. The final concentration of ethanol was less than 1% [25]. To avoid evaporation, filter paper was used to cover the plates [10]. Each test was consisted of 5 concentrations with four replicates. Rotenone (Aladdin, Shanghai, China) and H_2_O containing ethanol (1%) were employed as positive and negative controls, respectively. Both control and treated juveniles were incubated in the growth chamber at 27–29 °C in darkness. After treatment for 72 h, the mortality rate was determined. Juveniles were considered to be dead if they had no movement after stimulation with a fine needle.

### 3.4. EO Isolation and Fractionation

The crude EO of *Seseli mairei H. Wolff* roots (20 mL) was used for silica gel (900 g, 200 mesh, Qingdao Marine Chemical Co., Ltd., Qingdao, China) column chromatography (900 mm length, 80 mm i.d.) by gradient elution of ethyl acetate/petroleum ether (0:100, 1:100, 2:100, ∙∙∙, to 100:0). The fractions (600 mL) were harvested and evaporated at 40 °C, and 9 fractions were produced by combining related fractions based on their thin layer chromatography (TLC) profiles. The screening experiments were carried out by using 0.1% (*v/v*) solution of different fractions. Fractions (F3, F5, F8–F9) that exhibited nematicidal toxicity, with the same TLC profiles, were pooled and separated by preparative TLC until the three active compounds: (*E*)-2-decenal, falcarinol, and octanoic acid were obtained (Figure 1). The structures of the components were examined according to nuclear magnetic resonance. ^13^C and ^1^H-NMR spectra were determined with CDCl_3_ as a solvent using the Bruker AMX**50**0 [500 MHz (^1^H)] and ACF300 [300 MHz (^1^H)] instruments (Billerica, MA, USA). Tetramethylsilane was used as an internal standard.

### 3.5. GC-MS Analysis

GC-MS analysis was conducted using an Agilent system containing an Agilent Chem Station, a model 5973N mass selective detector, and a model 6890N gas chromatograph (Agilent Technologies Inc, City of Santa Clara, USA). The GC column was HP-5ms Capillary GC column [poly-(5%-diphenyl/95%-dimethylsiloxane), 30 m × 0.25 mm, 0.25 μm film thickness]. The oven temperature was initiated at 60 °C for 60 s, increased at 10 °C/min to 180 °C for 60 s, and further increased at 20 °C/min to 280 °C for 15 min. The injector temperature was kept at 270 °C. The sample (1 μL,) was diluted with acetone (1:100), and then injected at a ratio of 1:10. Helium was employed as the carrier gas, and the flow rate was 1.0 mL/min. Spectral scanning was conducted within the range of 20–550 m/z at 2 scans/s. The EO components were identified by comparing their mass spectra and retention index according to previous literature [32,33] and presented in NIST 98 as well as those of authentic compounds prepared in our laboratory. A homologues series of n-alkanes (C_8_–C_24_) were employed as reference points for calculating retention indices. Component relative percentage was determined according to GC peak areas without using the correction factors.

### 3.6. Data Analysis

Abbott’s formula was used to correct the mortality data. To calculate LC_50_ values and their 95% confidence intervals, Probit analysis was conducted on the obtained results using the PriProbit Program version 1.6.3 (Kyoto University, Kyoto, Japan) [34].

#### Appendix

Octanoic acid **(1,**
Figure 1**)**. Colorless oil, C_8_H_16_O_2_, ^1^H-NMR (500 MHz, CDCl_3_) *δ* (ppm): 0.88 (t, 3H, ^3^*J* = 7.1 Hz, 1-CH_3_), 1.21–1.38 (m, 11H), 1.59–1.67 (m, 2H, H-6), 2.32–2.37 (m, 2H, 7-H). ^13^C-NMR (125 MHz, CDCl_3_) *δ* 14.05 (C-1), 22.58 (C-2), 24.90 (C-6), 29.05* (C-4), 29.08* (C-5), 31.64 (C-3), 34.38 (C-7), 176.95 (C-8).

(*E*)-2-Decenal **(2,**
Figure 1**)**. Colorless oil, C_10_H_16_O, ^1^H-NMR (500 MHz, CDCl_3_) δ (ppm): 0.89 (3H, t, ^3^*J* = 7.8 Hz), 1.20–1.38 (8H, m), 1.51 (2H, p, ^3^*J* = 7.5 Hz), 2.34 (2H, ddt, ^3^*J* = 7.5 Hz, ^3^*J* = 7.0 Hz, ^3^*J* = 1.5 Hz), 6.12 (1H, ddt, ^3^*J* = 15.5 Hz, ^3^*J* = 7.0 Hz, ^4^*J* = 1.5 Hz), 6.86 (1H, dt, ^3^*J* =15.5 Hz, ^3^*J* = 7.0 Hz), 9.15 (1H, d, ^3^*J* = 7 Hz). ^13^C-NMR (125 MHz, CDCl_3_) *δ* 14.1 (C-10), 22.60 (C-9), 27.90 (C-8), 29.00* (C-7), 29.10* (C-6), 31.70 (C-5), 32.70 (C-4), 194.10 (C-1), 159.00 (C-2), 133.00 (C-3).

Falcarinol **(3,**
Figure 1**).** Yellow oil, ^1^H-NMR (500, MHz CDCl_3_) δ (ppm): 0.90 (3H, t, ^3^*J* = 6.9 Hz, H-17), 1.29–1.31 (8H, m, H-13, 14, 15, 16), 1.36–1.40 (2H, m, H-12), 2.04 (1H, s, H-11), 2.06 (1H, d, ^3^*J* = 7.8 Hz, H-3), 3.06 (2H, dd, ^3^*J* = 6.9, 0.7 Hz, H-8), 4.94 (1H, d, ^3^*J* = 5.3 Hz, H-3), 5.24- 5.28 (1H, m, H-1), 5.40 (1H, d, ^3^*J* = 10.5 Hz, H-9), 5.47 (1H, s, H-1), 5.50–5.53 (^1^H, m, H-10), 5.96 (1H, ddd, ^3^*J* = 17.0, 10.1, 5.4 Hz, H-10). ^13^C-NMR (CDCl_3_, 125 MHz) *δ* (ppm): 136.10 (C-2), 133.10 (C-10), 121.90 (C-9), 117.10 (C-1), 80.30 (C-7), 74.20 (C-4), 71.30 (C-5), 64.00 (C-6), 63.60 (C-3), 31.80 (C-15), 29.20 (C-12, 13, 14), 27.20 (C-11), 22.60 (C-16), 17.70 (C-8), 14.10 (C-17).

## Figures and Tables

**Figure 1 molecules-28-02205-f001:**
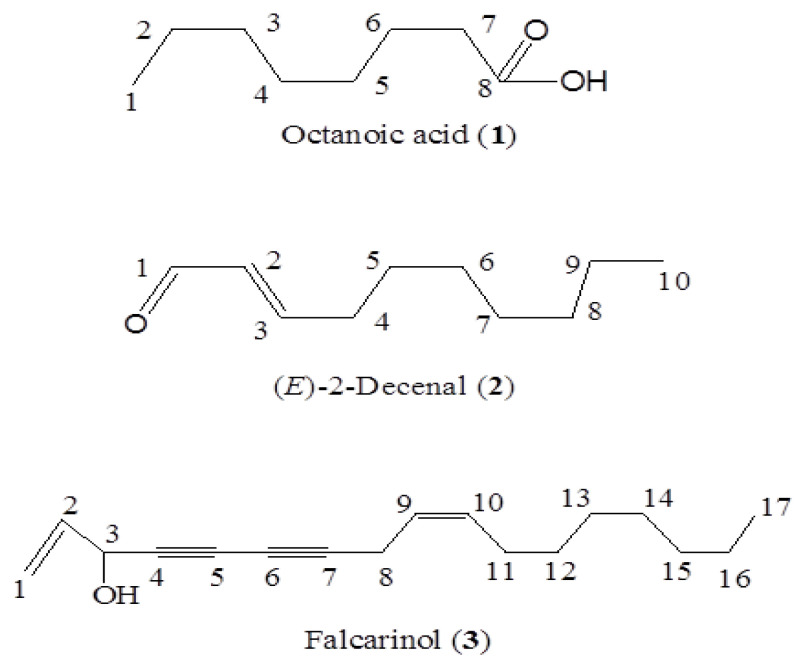
Structure of the extracted compounds.

**Table 1 molecules-28-02205-t001:** Chemical components of the EO isolated from *Seseli mairei* H. Wolff roots.

Peak	Compounds	RI *	Percentage (%)
	Monoterpenoids		27.50
1	α-Pinene	939	0.78
2	Camphene	954	1.53
3	Sabinene	975	2.87
4	β-Pinene	981	1.49
5	β-Myrcene	991	3.37
6	α-Phellandrene	1005	1.08
7	δ-3-Carene	1008	0.44
8	Limonene	1029	2.09
9	β-Phellandrene	1030	2.37
10	1,8-Cineole	1033	0.95
11	(*E*)-β-Ocimene	1048	1.35
12	γ-Terpinene	1057	2.05
13	Fenchone	1088	0.79
14	Linalool	1097	0.15
15	Camphor	1143	0.38
16	Borneol	1174	1.21
17	Terpineol-4-ol	1179	0.80
18	α-Terpineol	1191	3.16
19	Bornyl acetate	1287	0.64
	Sesquiterpenoids		23.73
20	δ-Elemene	1335	1.75
21	β-Cubebene	1387	0.65
22	β-Caryophyllene	1420	6.64
23	(*Z*)-β-Farnesene	1438	0.69
24	γ-Selinene	1475	0.61
25	Germacrene D	1485	4.28
26	β-Bisabolene	1506	0.59
27	Cuparene	1511	3.52
28	Spathulenol	1578	1.31
29	Caryophyllene oxide	1584	3.69
	Phenylpropanoids		3.14
30	Eugenol	1351	0.78
31	Methyleugenol	1403	1.41
32	Elemicin	1558	0.95
	Others		43.47
33	Hexanoic acid	987	0.87
34	Octanoic acid	1172	10.49
35	(*E*)-2-Decenal	1265	6.17
36	Senkyunolide	1729	0.69
37	Falcarinol	2038	25.25
	Total		97.84

* RI, retention index as determined on a HP-5MS column using the homologous series of *n*-alcane.

**Table 2 molecules-28-02205-t002:** The nematicidal toxicity of the essential oil of *Seseli mairei* H. Wolff roots against *Bursaphelenchus xylophilus*.

Treatments	LC_50_ (μg/mL)	95% FL *	Slope ± SE	Chi Square (χ^2^)
Essential oil	53.45	48.61–57.28	2.25 ± 0.21	13.08
(*E*)-2-Decenal	176.34	159.89–193.21	1.21 ± 0.11	8.96
Falcarinol	8.53	7.76–9.41	1.83 ± 0.17	10.32
Octanoic acid	65.56	59.44–72.35	2.45 ± 0.24	7.51
Rotenone	2.32	1.96–2.51	4.25 ± 0.38	6.76

FL *: Fiducial limits.

## Data Availability

The data presented in this study are available on request from the corresponding authors.

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
