# Peer review of "GC-MS Analysis of the Essential Oil from Seseli mairei H. Wolff (Apiaceae) Roots and Their Nematicidal Activity"

_molecules, 2023, doi:10.3390/molecules28052205_

Round 1

Reviewer 1 Report

Dear authors,

Please consider the following aspects:

1. The correct systematic name of the plant is Seseli mairei H. Wolff (this name must be used throughout the paper)

2. All Latin systematic names must be written systematically correct and italics need to be used. It is about the name of your plant, plants/essential oils you are comparing with, or the Latin name of the nematodes. Correct writing is random throughout the paper.

3. According to the International System, the liter and submultiples of the liter are written: L, mL, uL. Please correct

4. For a better understanding, in the material and methods section, it would be desirable to group the materials that address the essential oil, as well as those that refer to nematodes. So 2.1+2.4+2.5, followed by 2.2 and 2.3, then 2.6.

5. Please review the English language. E.g. Row 37: "Therefore, pesticides with less toxic needed to be developed". Maybe less toxic activity

Reviewer 2 Report

This is an interesting manuscript. I have suggested some changes and also left some questions that can improve the quality of this material.

Lines 9-10: Rephrase it. It’s not making sense in the way that is written.

Line 21: Were suggested?

Line 23: I would suggest not using Essential oil on your keyword, since you already have it in the title.

Lines 26-27: Italic

Line 37: With less toxic what?

Line 41: Someone reviewed, not the literature.

Line 41: ‘such as plants in’ does not make sense.

Lines 41-43: Italic

Line 45: Common name is not required unless it is in English.

Line 56: Is it possible to get information about collection (Date, location etc)?

Lines 61-62: Is this steam-distillation or hydro-distillation?

Lines 61-62: Name of the equipament?

Lines 85-86: Why x10 and x5?

Line 88: Can we call it a fumigation assay?

Line 126: Why Abbott?

Line 152: What is the criteria to the determine what is a major components? Why 37?

Lines 157-158: It is not 37 anymore?

Line 164: Italic.

Lines 167-175: Italic missing all over the place.

It would be great if you add why would be the reason for such variation in the essential oils’ composition, instead of just extensively compare the composition. This worth to be mentioned.

Lines 176-186: Italic.
Line 177: μg/ml instead of g/ml

Lines 182-185: You can’t make this comparison. It’s not the same unit. You will have to adjust the unit based on the density of the essential oil used in your study.

Line 191: I wouldn’t call weaker. It was just less toxic, not weaker.

Line 198: Considering that..

Round 2

Reviewer 2 Report

The manuscript was well improved. I just have one more suggestion on the first paragraph of the discussion. Please write it as 37 components instead of 37 major components.

Author Response

Thank you so much for your supervision of the reviewing process of our manuscript with the reference number of 2190822. Since you and the reviewers found the topic of interest, we are encouraged greatly. According to your comments, we have carefully revised our manuscript point by point. We use a professional agency to revise and improve the English language and to reduce the repetition rate of the manuscript. We hope that these revisions are satisfactory and that the revised version can meet the qualification for publication in Molecules.
